# Partial Label Unsupervised Domain Adaptation with Class-Prototype Alignment

**Yan Yan**[1], **Yuhong Guo**[1,2]
[1]Carleton University, Ottawa, Canada      [2]CIFAR AI Chair, Amii, Canada
`yanyan@cunet.carleton.ca, yuhong.guo@carleton.ca`

## Abstract

Partial label learning (PLL) tackles the problem where each instance is associated with a set of candidate labels, only one of which is the ground-truth label. Most existing PLL approaches assume that both the training and test sets share an identical data distribution. However, this assumption does not hold in many real-world scenarios where the training and test data come from different distributions. In this paper, we formalize this learning scenario as a new problem called partial label unsupervised domain adaptation (PLUDA). To address this challenging PLUDA problem, we propose a novel Prototype Alignment based PLUDA method named PAPLUDA, which dynamically refines the pseudo-labels of instances from both the source and target domains by consulting the outputs of a teacher-student model in a moving-average manner, and bridges the cross-domain discrepancy through inter-domain class-prototype alignment. In addition, a teacher-student model based contrastive regularization is deployed to enhance prediction stability and hence improve the class-prototypes in both domains for PLUDA. Comprehensive experimental results demonstrate that PAPLUDA achieves state-of-the-art performance on the widely used benchmark datasets.

## 1 Introduction

Partial label learning (PLL) is a typical weakly supervised learning problem, where each training instance is assigned a candidate label set, only one of which is valid. PLL has gained increasing attention from the research community due to its effectiveness in reducing annotation costs in various real-world scenarios, such as face naming (Hüllermeier & Beringer, 2006), web mining (Luo & Orabona, 2010), and ecoinformatics (Liu & Dietterich, 2014). Nevertheless, standard PLL assumes the training and test data are sampled from the same distribution. With this assumption, a model learned from the training data is expected to generalize well on the test data. However, this assumption does not hold in many real-world scenarios where the training and test data come from different distributions—e.g., the training and test data are collected from different sources, or we have an outdated training set due to the fact that data always change over time. In such cases, there would be a discrepancy between the training and test data distributions, and hence naively adopting the off-the-shelf PLL models can lead to significant test performance degradation. Meanwhile, the unavailability of the ground-truth labels prevents the deployment of existing unsupervised domain adaptation (UDA) methods (Tzeng et al., 2017; Dong et al., 2021; Na et al., 2021; Shen et al., 2022).

We formalize this new learning scenario of PLL with training-test distribution gaps as a partial label unsupervised domain adaptation (PLUDA) problem. By integrating the challenges of both PLL and UDA problems, the PLUDA problem has the following characteristics: (1) the source and target domains have different distributions but share the same set of classes; (2) data in the source domain have only partial labels—each instance is associated with a candidate label set, while the target domain only has unlabeled data; (3) the candidate label set for each source instance can contain both the ground-truth and irrelevant noisy labels, while labels outside of the candidate set are true negative labels. The goal of the PLUDA task is to learn a domain-invariant prediction model from the partial-label source domain that can generalize well in the unlabeled target domain.

Although both PLL and UDA have been studied intensively in the literature, to the best of our knowledge, there is no research yet to address the integrated challenges of PLUDA in a unified

framework. PLUDA is related to but still distinct from the weakly supervised domain adaptation (WSDA) problem studied in the recent literature (Shu et al., 2019; Xie et al., 2022). WSDA assumes the ground-truth labels of the source domain instances are corrupted (e.g., replaced) with noisy labels, and has been studied as an effort of reducing annotation costs. Some researchers address the WSDA problem based on off-the-shelf technologies and obtain good performance. For example, Xie et al. (2022) exploit the bilateral relationships between the source and target domains to construct a universal framework, GearNet, based on the existing domain adaptation methods of TCL (Shu et al., 2019) and DANN (Tzeng et al., 2017).

In this paper, we propose a novel prototype alignment based partial label unsupervised domain adaptation approach, PAPLUDA, to address the combined PLL and UDA problems simultaneously in a newly formalized PLUDA learning scenario. The proposed PAPLUDA approach contains three pseudo-label based components that collaborate with each other to tackle PLUDA learning. First, we conduct soft label disambiguation to dynamically rectify the pseudo-labels of training instances from both domains toward the ground-truth labels, which aims to disambiguate the partial labels and set up the foundation for cross-domain class alignment. Second, we propose an inter-domain class-prototype based alignment to minimize the discrepancy between the same class prototypes from the source and target domains while maximizing the gaps between the prototypes of different classes. Finally, we deploy a teacher-student model based contrastive regularization to enhance the reliable pseudo-labels, and hence improve the class-prototypes and the inter-domain prototype alignment. Overall, the contributions of this paper can be summarized as follows:

- A new challenging learning problem, PLUDA, is proposed, which is more practical than separate PLL and UDA problems by simultaneously dropping the common supervised learning assumptions of accurate data labels and identical training and testing distributions.
- A novel PAPLUDA approach is proposed to tackle the PLUDA problem.
- Comprehensive experiments are conducted on benchmarks and the results validate the effectiveness of the proposed PAPLUDA.

## 2    RELATED WORK

### 2.1    PARTIAL LABEL LEARNING

Partial label learning is a prevalent weakly supervised learning problem (Zhou, 2018), where each training instance is associated with a set of candidate labels, only one of which is valid. Many methods adapt existing learning techniques to the partial-label training data and disambiguate the candidate noisy labels by aggregating predictions. For maximum likelihood techniques, the likelihood of each partial label training instance is produced by consulting the probability of each candidate label associated with the training instance (Jin & Ghahramani, 2002; Liu & Dietterich, 2012). For $k$-nearest neighbor techniques, the candidate labels from neighbor instances are integrated for final prediction in a weighted voting manner (Hüllermeier & Beringer, 2006; Gong et al., 2017; Zhang & Yu, 2015). For maximum margin techniques, the classification margin on each partial label instance is defined by discriminating the modeling outputs from candidate labels and non-candidate labels (Nguyen & Caruana, 2008; Yu & Zhang, 2016). Apart from adapting the off-the-shelf techniques to partial-label data, some researchers propose to address PLL by adapting the partial-label data to existing learning techniques. For example, Zhang et al. (2017) propose to transform a partial-label training set into multiple binary training sets which can then be used to built multiple binary classifiers corresponding to the ECOC coding matrix. Wu & Zhang (2018) adopt a one-vs-one decomposition strategy to enable binary decomposition for learning from partial-label data. Although these works produce competitive performance, they are restricted to linear models and have difficulty to handle large-scale datasets.

To alleviate the limitations of the standard PLL methods, deep learning based PLL has recently started gaining attention from the research community. Yao et al. (2020) attempt to address PLL with deep convolutional neural networks by exploiting a temporal-ensembling technique. Meanwhile, Yan & Guo (2020) handle PLL with multilayer perceptrons by means of batch label correction. Wen et al. (2021) present a group of loss functions called leveraged weighed loss, which takes the work of (Lv et al., 2020) as its special case. Feng et al. (2020) present two provably consistent methods from the perspective of partial label generation: a risk-consistent method and

a classifier-consistent method. Xu et al. (2021) adopt variational inference along network training to progressively refine the latent label distributions, assuming an instance-dependent partial label generation process. In addition, Wang et al. (2022) propose a class prototype based label disambiguation method through contrastive learning, which achieves impressive performance on several image classification benchmarks. Although theses deep learning methods have produced notable progress on PLL, the study however is still limited to the supervised learning scenario where both training and test datasets share an identical data distribution.

## 2.2 Unsupervised Domain Adaptation

Unsupervised domain adaptation (UDA), which aims to transfer knowledge learned from a label-rich source domain to an unlabeled target domain, has gained tremendous attention from the research community (Hoffman et al., 2018; Ghafoorian et al., 2017; Kamnitsas et al., 2017; Wang & Zheng, 2015; Fang et al., 2020; Dong et al., 2021). The main challenge of UDA is to bridge the cross-domain distribution gap and induce transferable prediction models. Some previous research works have attempted to minimize the inter-domain gap based on standard distribution discrepancy criteria, such as the maximum mean discrepancy (Pan et al., 2010), Kullback-Leibler divergence (Zhuang et al., 2015), central moment discrepancy (Zellinger et al., 2017), and Wasserstein distance (Lee & Raginsky, 2018). In addition, several studies attempt to minimize the domain discrepancy by exploiting a domain discriminator in an adversarial learning manner, including domain adversarial network (Ganin et al., 2016) and adversarial discriminative domain adaptation (Tzeng et al., 2017). Recently, some researchers consider enhancing the domain adaptation procedure by exploiting the unlabeled data from the target domain, and propose new UDA approaches based on self-training models (Zou et al., 2019; Chen et al., 2020b).

These standard UDA methods however all assume the training data in the source domain are annotated with accurate true labels, which are difficult to provide in real-world scenarios. More recently, weakly supervised domain adaptation (WSDA), which addresses both the UDA problem and the noisy label learning issue, has gained increasing attention in reducing annotation demands for high-quality labels (Shu et al., 2019; Tzeng et al., 2017; Liu et al., 2019). For example, TCL (Shu et al., 2019) proposes to train a domain adaptation network by exploiting the selected clean and transferable source instances. GearNet (Xie et al., 2022) proposes a universal framework based on the off-the-shelf powerful domain adaptation methods, TCL (Shu et al., 2019) and DANN (Tzeng et al., 2017), and exploits the bilateral relationships between the source and target domains to enhance the WSDA training procedure. In this work, we relax the label requirement in the source domain by considering partial labels, which can be easily obtained with crowdsourcing. This new PLUDA problem cannot be directly addressed by existing PLL or UDA methods.

## 3 Proposed Approach

We assume a source domain $D_S = \{(\mathbf{x}_i^s, \mathbf{y}_i^s)\}_{i=1}^{n_s}$ with $n_s$ partial-label instances and a target domain $D_T = \{\mathbf{x}_i^t\}_{i=1}^{n_t}$ with $n_t$ unlabeled instances, where $\mathbf{x}_i^s$ and $\mathbf{x}_i^t$ denote the $i$-th instance from the source domain $D_S$ and the target domain $D_T$ respectively, and $\mathbf{y}_i^s \in \{0,1\}^L$ denotes the label indicator vector associated with $\mathbf{x}_i^s$. $L$ denotes the number of classes. The multiple 1 values in $\mathbf{y}_i^s$ denote either the true label or irrelevant label noise indistinguishably, which forms the candidate label set $S_i \subseteq \{1, \cdots, L\}$ for $\mathbf{x}_i^s$. We use $B_s = \{(\mathbf{x}_i^s, \mathbf{y}_i^s)\}_{i=1}^{m_s}$ and $B_t = \{\mathbf{x}_i^t\}_{i=1}^{m_t}$ to denote a mini-batch of instances sampled from $D_S$ and $D_T$, respectively. The task of PLUDA is to induce a multi-class prediction model from $D_S$ and $D_T$ that can generalize well in the target domain.

We propose to address this novel task of PLUDA by simultaneously performing domain adaptation and partial label disambiguation with inter-domain class-prototype alignment and teacher-student model based soft label disambiguation. The proposed PAPLUDA model is illustrated in Figure 1. The model architecture includes a shared feature extractor $G$, a teacher-student prediction network (student network $F$ and teacher network $F'$), and a feature projection network $H$. Towards the objective of PAPLUDA, three loss components are designed on the model architecture: (1) *classification loss with soft label disambiguation*, which trains a teacher-student model from the partial-label data in the source domain by performing moving average based soft label disambiguation; (2) *inter-domain class-prototype alignment loss*, which aims to bridge the cross-domain distribution gap for information sharing by performing contrastive alignment between the prototypes of the same class

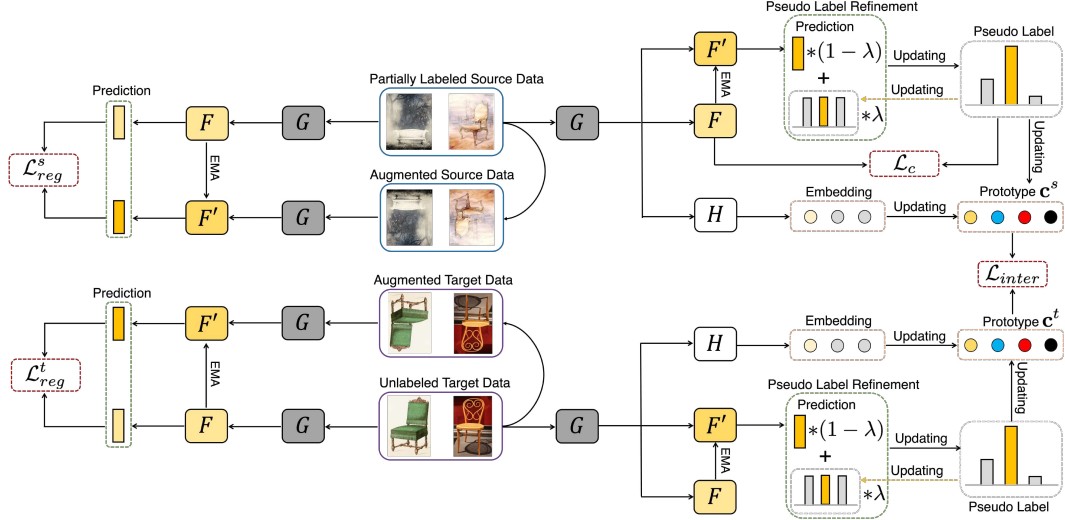

Figure 1: The proposed PAPLUDA model, which mainly has three loss components: classification loss with soft label disambiguation $\mathcal{L}_c$; inter-domain class-prototype alignment loss $\mathcal{L}_{inter}$; teacher-student model based contrastive regularization loss $\mathcal{L}_{reg}$ ($\mathcal{L}_{reg}^s$ and $\mathcal{L}_{reg}^t$). All the components mutually benefit from each other to improve the PLUDA learning cooperatively.

in the two domains; and (3) *teacher-student model based contrastive regularization loss*, which enhances prediction stability and consequently improves class-prototypes and inter-domain prototype alignment for PLUDA learning. We elaborate these loss components and the proposed approach in the following subsections.

## 3.1 CLASSIFICATION WITH SOFT LABEL DISAMBIGUATION

As the true labels are hidden within each candidate label set in the source training data, it is critical to perform label disambiguation—that is, separate the true labels from the noisy labels—for inducing a good classification model. We deploy a teacher-student model as the classification model due to its robustness against noisy labels for stable prediction (Li et al., 2019). We assume the teacher and the student share the same feature extractor $G$ but maintain two different prediction networks: student network $F$ and teacher network $F'$. $F$ and $F'$ have identical structures but different parameters, and they produce multi-class softmax probability prediction vectors. The parameters of the student network, $\Theta_F$, will be learned from the training data, while the parameters of the teacher network, $\Theta_{F'}$, will be updated as an exponential moving average (EMA) of the student network parameters (Tarvainen & Valpola, 2017), such as:

$$\Theta_{F'} = \eta\Theta_{F'} + (1-\eta)\Theta_F, \tag{1}$$

where $\eta \in (0,1)$ is a weight hyperparameter. Importantly, we propose to use the teacher network that is typically more robust than the student network to perform label disambiguation:

$$\widehat{\mathbf{y}}_{ij}^s = \mathbb{I}\big[j = \arg\max_{j' \in S_i} F'(G(\mathbf{x}_i^s))\big], \tag{2}$$

where $\mathbb{I}$ is an indicator function, and $\widehat{\mathbf{y}}_i^s$ is the disambiguated one-hot label indicator vector for instance $\mathbf{x}_i^s$ using the teacher network. Such disambiguated one-hot label vectors for the source training instances however would certainly contain mistakes. Directly using them for model training could cause oscillations by largely deviating from the original candidate labels. Hence we propose to perform *soft label disambiguation* by updating the candidate label distribution vectors with the one-hot vectors $\{\widehat{\mathbf{y}}_i^s\}$ in a moving average fashion:

$$\mathbf{p}_i^s = \lambda\mathbf{p}_i^s + (1-\lambda)\widehat{\mathbf{y}}_i^s, \tag{3}$$

where $\lambda \in (0, 1)$ is a hyperparameter, and $\mathbf{p}_i^s$ denotes the soft disambiguated label vector for the source instance $\mathbf{x}_i^s$. We initialize $\mathbf{p}_i^s$ as the original candidate label distribution vector for $\mathbf{x}_i^s$, such as $\mathbf{p}_{ij}^s = \frac{1}{|S_i|}\mathbb{I}[j \in S_i]$.

Benefiting from the moving average update, we expect this soft label disambiguation procedure can steadily and progressively approach the underlying true label vectors through the iterative training steps. With the disambiguated soft label vectors, we can update the student network $F$ and the feature extractor $G$ by minimizing following classification loss for each instance:

$$\mathcal{L}_c^i(\mathbf{x}_i^s, S_i) = \sum\nolimits_{j=1}^{L} -\mathbf{p}_{ij} \log F_j(G(\mathbf{x}_i^s)) \tag{4}$$

where $j$ denotes the indices of the class labels, and $F_j(\cdot)$ denotes the $j$-th output entry of the student prediction network $F$, which indicates the predicted probability of the given instance belonging to the $j$-th class. By iteratively performing soft label disambiguation and prediction model update, we expect the true labels can be gradually and statistically identified from the source training data, and hence induce a good classification model.

## 3.2 INTER-DOMAIN CLASS-PROTOTYPE ALIGNMENT

The goal of PLUDA is to induce a classification model that generalizes well in the target domain. Hence it is important to take the unlabeled target data into consideration and bridge the cross-domain feature distribution discrepancy. In particular, we propose to discriminatively bridge the cross-domain divergence and match the two domains specifically for the given classification task by performing prediction-aware inter-domain class-prototype alignment.

A class-prototype is defined as the mean embedding vector of the instances belonging to the class. Following self-supervised learning (Chen et al., 2020a), we compute the class-prototypes in a low-dimensional embedding space produced by a projection network $H$ on the extracted features. As the true labels are unknown in the source and target domains, we propose to calculate the class-prototypes based on the pseudo-labels predicted by the teacher-student model. Specifically, the class-prototypes for each training batch $B_d$ are computed as follows:

$$\mathbf{c}_k^d = \frac{\sum_{\mathbf{x}_i^d \in B_d} H(G(\mathbf{x}_i^d)) \cdot \mathbb{I}[k = \arg\max_{k' \in \mathcal{Y}^d} \mathbf{p}_{ik'}^d]}{\sum_{\mathbf{x}_i^d \in B_d} \mathbb{I}[k = \arg\max_{k' \in \mathcal{Y}^d} \mathbf{p}_{ik'}^d]} \tag{5}$$

where $d \in \{s, t\}$ is a domain indicator; $\mathcal{Y}^s = S_i$ in the source domain and $\mathcal{Y}^t = \{1, 2, \cdots, L\}$ in the target domain. The discrete pseudo-labels in the source domain are produced from the disambiguated soft label vectors $\{\mathbf{p}_i^s\}$ computed in Eq.(3). For the unlabeled target domain, we propose to maintain soft label probability vectors $\{\mathbf{p}_i^t\}$ during training in the following way:

$$\mathbf{p}_i^t = \lambda \mathbf{p}_i^t + (1 - \lambda) F'(G(\mathbf{x}_i^t)). \tag{6}$$

To alleviate the negative impact and limitation of computing class-prototypes using the partial-label data in a local batch, we maintain a dynamic dictionary $C'^s = [\mathbf{c}_1'^s, \cdots, \mathbf{c}_k'^s, \cdots, \mathbf{c}_L'^s]$ in the source domain to store the class-prototypes from the previous batch for prototype calibration. After computing the class-prototypes using Eq.(5) for the current source batch, we further update them by consulting the dynamic dictionary in a moving average manner:

$$\mathbf{c}_k^s = \gamma \mathbf{c}_k^s + (1 - \gamma) \mathbf{c}_k'^s, \tag{7}$$

where $\gamma \in [0, 1]$ is a trade-off parameter. This allows us to take the prototype information from previous batches into account, overcoming the potential random bias in a local batch.

With the class-prototypes computed in both domains, we propose to perform inter-domain prototype alignment using the following contrastive loss for each class $k$:

$$\mathcal{L}_{inter}^k(\mathbf{c}_k^s, \mathbf{c}_k^t) = -\log \frac{\Omega(\mathbf{c}_k^s, \mathbf{c}_k^t)}{\Omega(\mathbf{c}_k^s, \mathbf{c}_k^t) + \sum_{j=1}^{L} \mathbb{I}[j \neq k] \cdot \left(\Omega(\mathbf{c}_j^s, \mathbf{c}_k^t) + \Omega(\mathbf{c}_j^t, \mathbf{c}_k^s)\right)}, \tag{8}$$

where $\Omega(a, b) = \exp(\text{cosine}(a, b)/\tau)$ measures the similarity of the given pair, and $\tau$ is the temperature hyperparameter. This contrastive loss uses the inter-domain class-prototype pair from the same

class as the positive sample and the pairs from different classes as the negative samples. We expect this contrastive loss can push the inter-domain prototype pair of the same class closer to each other and push the inter-domain pair from different classes apart in the embedded feature space, aiming to induce a good feature representation space, in which the two domains can be discriminatively aligned for classification model training.

### 3.3 TEACHER-STUDENT MODEL BASED CONTRASTIVE REGULARIZATION

To exploit the consistent statistical knowledge learned along model training across both domains and avoid prediction perturbations caused by the noisy label information, we propose a self-supervised contrastive loss in the output space of the prediction networks and calculate the loss in both domains as a regularizer to enhance the stability of the prediction model.

The self-supervised contrastive loss aims to push the prediction outputs for the variants of the same instance to be similar and for different instances to be dissimilar. It can conveniently exploit the unlabeled target instances. Specifically, for each training instance $\mathbf{x}_i$, we first generate an augmented version of it as $\widehat{\mathbf{x}}_i = \mathrm{Aug}(\mathbf{x}_i)$, where $\mathrm{Aug}(\cdot)$ denotes the data augmentation function (Cubuk et al., 2020). Then the contrastive loss for each instance $\mathbf{x}_i^d$ in a batch $B_d$ ($d \in \{s, t\}$) can be computed as:

$$\mathcal{L}_{reg}^{i,d}(\mathbf{x}_i^d, \widehat{\mathbf{x}}_i^d) = -\log \frac{h(\mathbf{x}_i^d, \widehat{\mathbf{x}}_i^d)}{\sum_{\mathbf{x}_j^d \in B_d} h(\mathbf{x}_j^d, \widehat{\mathbf{x}}_i^d) + \sum_{\mathbf{x}_j^d \in B_d} \mathbb{I}[j \neq i] \cdot h(\widehat{\mathbf{x}}_j^d, \widehat{\mathbf{x}}_i^d)}, \tag{9}$$

where $h(a, b) = \exp(\mathrm{cosine}(F(G(a)), F'(G(b)))/\tau)$ denotes the similarity between the predicted label vectors on the given input instance pair using the student network and teacher network respectively, and $\tau$ is the temperature hyperparameter. By deploying this contrastive loss as a regularizer on both domains, we have the following batch-wise total regularization loss:

$$\mathcal{L}_{reg} = \mathbb{E}_{\mathbf{x}_i^s \in B_s} \mathcal{L}_{reg}^{i,s}(\mathbf{x}_i^s, \widehat{\mathbf{x}}_i^s) + \mathbb{E}_{\mathbf{x}_i^t \in B_t} \mathcal{L}_{reg}^{i,t}(\mathbf{x}_i^t, \widehat{\mathbf{x}}_i^t) \tag{10}$$

We expect this prediction consistency regularization loss can help induce similar feature representations across domains regarding the classification task and improve the inter-domain class-prototype alignment to facilitate domain adaptation.

### 3.4 OVERALL TRAINING LOSS

Finally, by integrating the classification loss in Eq.(4), the inter-domain class-prototype alignment loss in Eq.(8) and the teacher-student model based contrastive regularization loss in Eq.(10) together, we have the following overall batch-wise training loss:

$$\mathcal{L} = \mathbb{E}_{\mathbf{x}_i^s \in B_s} \mathcal{L}_c^i + \alpha \sum_k \mathcal{L}_{inter}^k + \beta \mathcal{L}_{reg} \tag{11}$$

where $\alpha$ and $\beta$ are trade-off hyperparameters that balance the weights of the inter-domain class-prototype alignment loss and the teacher-student model based contrastive regularization loss respectively. The training algorithm is presented in Algorithm 1 in Appendix.

## 4 EXPERIMENTS

To validate the efficacy of the proposed PAPLUDA, we conduct experiments on two domain adaptation benchmark datasets by generating partial-label data in the source domain with different noise levels. We report our experimental setting and results in this section.

### 4.1 EXPERIMENT SETTING

**Datasets** We conduct experiments on two widely used domain adaptation datasets: Office-31 (Saenko et al., 2010) and Office-Home (Venkateswara et al., 2017). Office-31 is a classical dataset consisting of 4,652 images from 31 classes, which is distributed across the following three domains: Amazon (A), Webcam (W), and DSLR (D). These domains contain images collected from amazon.com, web camera, and digital SLR camera, respectively. Six different domain adaptation tasks can be constructed from this dataset, each of which uses one domain as the source domain and

Table 1: Test accuracy (mean±std, %) comparison on the partial-label Office-31 dataset with label ambiguity level $q = 0.2$. The best result in each column is highlighted in bold.

| Tasks | A → W | A → D | W → A | W → D | D → A | D → W | Average |
|---|---|---|---|---|---|---|---|
| DANN | 51.43±1.71 | 55.20±1.01 | 44.28±0.72 | 72.08±1.13 | 38.10±1.41 | 64.09±0.45 | 54.20±1.07 |
| TCL | 61.72±0.88 | 68.54±1.21 | 47.58±0.36 | 71.66±1.05 | 42.15±1.42 | 58.46±0.61 | 58.35±0.92 |
| GearNet$_{Co-teach.}$ | 33.20±1.97 | 39.37±1.28 | 38.99±0.68 | 63.37±2.08 | 33.80±1.51 | 52.24±1.85 | 43.50±1.56 |
| GearNet$_{DANN}$ | 62.76±0.48 | 64.79±1.05 | 46.37±1.19 | 78.88±1.62 | 39.71±1.02 | 67.08±0.81 | 59.93±1.03 |
| GearNet$_{TCL}$ | 63.55±0.82 | 69.85±1.03 | 49.24±1.23 | 73.13±0.91 | 43.86±0.99 | 60.44±0.66 | 60.01±0.94 |
| PAPLUDA(Ours) | **79.36±0.46** | **78.72±1.15** | **53.34±0.87** | **94.37±0.67** | **50.07±0.56** | **87.11±0.86** | **73.82±0.76** |

Table 2: Test accuracy (mean±std, %) comparison on the partial-label Office-31 dataset with label ambiguity level $q = 0.4$. The best result in each column is highlighted in bold.

| Tasks | A → W | A → D | W → A | W → D | D → A | D → W | Average |
|---|---|---|---|---|---|---|---|
| DANN | 40.36±1.59 | 40.01±1.28 | 35.62±1.65 | 57.08±1.62 | 30.61±1.92 | 52.73±1.81 | 42.73±1.64 |
| TCL | 49.08±1.59 | 61.45±1.88 | 39.95±0.93 | 61.25±1.79 | 35.61±1.87 | 55.11±2.57 | 50.40±1.77 |
| GearNet$_{Co-teach.}$ | 24.52±1.18 | 28.67±0.88 | 22.69±1.08 | 45.21±1.36 | 21.80±1.54 | 32.85±1.43 | 29.29±1.24 |
| GearNet$_{DANN}$ | 50.52±1.63 | 48.33±1.91 | 38.45±1.51 | 66.25±1.86 | 33.38±1.07 | 55.84±1.31 | 48.79±1.55 |
| GearNet$_{TCL}$ | 53.51±0.81 | 64.79±1.03 | 41.05±1.93 | 63.87±1.91 | 36.17±1.44 | 56.90±1.60 | 52.71±1.45 |
| PAPLUDA(Ours) | **76.62±1.06** | **77.15±1.01** | **51.25±0.98** | **91.56±1.66** | **45.28±1.12** | **82.42±1.35** | **70.71±1.19** |

uses another one as the target domain. Office-Home is a challenging domain adaptation benchmark dataset consisting of 15,500 images from 65 classes across the following four domains: Artistic (Ar), Clip Art (Cl), Product (Pr), and Real-World (Rw). These domains represent four different image styles, including artistic depictions, clipart images, images without background, and photos taken with cameras respectively. There are twelve different domain adaptation tasks constructed from Offce-Home, each of which uses a different ordered pair of domains as the source and target domains. For the PLUDA experiments, we produce partial labels in each source domain of the two datasets following the partial label generation process in (Lv et al., 2020). Specifically, each irrelevant label for an image is uniformly selected as a candidate label with a probability $q$, which controls the label ambiguity level of the produced dataset. When encountering a special case where no irrelevant label is chosen for an image, we randomly add an irrelevant label to the candidate label set to ensure the whole training set is corrupted thoroughly. For each comparison experiment, we report the average test accuracy and standard deviation based on five independent runs.

**Comparison Methods** We compare the proposed PAPLUDA approach with three state-of-the-art weakly supervised domain adaptation methods (GearNet$_{Co-teaching}$, GearNet$_{DANN}$, and GearNet$_{TCL}$) and two baseline methods (*DANN* and *TCL*). Each method is configured with the suggested parameters according to the respective literature. *DANN* (Tzeng et al., 2017) designs an adversarial domain adaptation mechanism for neural networks to reduce the cross-domain divergence. *TCL* (Shu et al., 2019) devises a transferable curriculum to select instances for model training, which addresses weakly supervised domain adaptation through self-paced learning. GearNet$_{Co-teaching}$ (Xie et al., 2022) exploits the bilateral relationship between source and target domains for model training and integrates the off-the-shelf co-teaching framework (Han et al., 2018) to improve the model's robustness against noisy labels. GearNet$_{DANN}$ (Xie et al., 2022) and GearNet$_{TCL}$ (Xie et al., 2022) are two universal frameworks for exploiting knowledge from both the source and target domains. They incorporate the effective domain adaptation methods, *DANN* and *TCL*, as their core components respectively to address the weakly supervised domain adaptation problem.

**Implementation Details** To ensure a fair comparison, we adopt the same backbone network and optimizer for all the comparison methods. We use ResNet50 as the backbone network for feature extraction. The projection network is a two-layer perceptron that outputs 128-dimensional embeddings. The prediction network is a two-layer perceptron followed by a softmax activation function. The coefficient hyperparameter $\eta$ for the teacher-student model and the hyperparameter $\gamma$ for class-prototype update are set to 0.999 and 0.99, respectively. For the soft label disambiguation, we linearly ramp down $\lambda$ from 0.95 to 0.8. The temperature parameter $\tau$ for similarity calculation is set to 5. We employ a standard stochastic gradient descent optimizer with a momentum of 0.9, a weight decay of 0.0005, an initial learning rate of 0.01, and a cosine learning rate decay. The mini-batch size is set to 64. The trade-off parameters $\alpha$ and $\beta$ are chosen from the set $\{0.1, 0.3, 0.5, 0.7, 0.8, 1\}$ for the two datasets according to the validation performance of the transfer task A → W on the Office-31 dataset and the Ar → CI task on the Office-Home dataset respectively.

Table 3: Test accuracy (mean±std, %) comparison on the partial-label Office-Home dataset with label ambiguity level $q = 0.2$. The best result in each row is highlighted in bold.

| Tasks | DANN | TCL | GearNet$_{Co-teach.}$ | GearNet$_{DANN}$ | GearNet$_{TCL}$ | PAPLUDA(Ours) |
|---|---|---|---|---|---|---|
| Ar → Cl | 17.03±1.16 | 21.89±1.35 | 16.47±0.62 | 18.18±1.10 | 22.44±0.81 | **34.14±0.87** |
| Ar → Pr | 29.66±1.17 | 32.45±1.37 | 25.33±0.76 | 33.33±1.17 | 33.99±1.01 | **46.78±0.97** |
| Ar → Rw | 38.28±2.06 | 45.59±0.75 | 38.19±0.72 | 42.14±1.37 | 47.81±1.62 | **55.68±1.33** |
| Cl → Ar | 22.66±0.75 | 31.71±1.58 | 18.22±0.73 | 27.42±0.78 | 33.49±1.58 | **40.76±0.94** |
| Cl → Pr | 27.69±1.25 | 34.57±1.29 | 28.80±0.86 | 33.08±1.16 | 35.19±1.06 | **49.24±1.26** |
| Cl → Rw | 31.09±1.45 | 40.57±1.43 | 27.21±0.88 | 37.52±0.64 | 42.34±0.93 | **52.35±0.81** |
| Pr → Ar | 22.46±0.61 | 33.87±0.62 | 19.17±1.38 | 26.88±0.91 | 34.33±1.35 | **43.25±0.84** |
| Pr → Cl | 17.25±1.83 | 23.98±1.83 | 14.28±1.25 | 18.87±1.22 | 25.11±1.44 | **39.60±1.12** |
| Pr → Rw | 37.22±1.06 | 50.43±0.87 | 38.24±0.79 | 42.53±1.08 | 54.41±1.22 | **65.31±1.05** |
| Rw → Ar | 31.17±1.12 | 46.33±1.01 | 26.29±1.05 | 35.88±0.72 | 47.45±1.56 | **57.34±0.74** |
| Rw → Cl | 21.32±0.84 | 28.58±1.89 | 18.76±1.20 | 22.25±1.01 | 30.97±1.46 | **43.22±1.14** |
| Rw → Pr | 47.96±1.52 | 57.74±1.21 | 48.68±1.23 | 52.97±1.35 | 62.54±1.21 | **72.30±1.13** |
| Average | 28.65±1.24 | 37.31±1.27 | 26.64±0.96 | 32.59±1.04 | 39.17±1.27 | **50.00±1.02** |

## 4.2 COMPARISON RESULTS

### 4.2.1 RESULTS ON OFFICE-31

The comparison results in terms of test accuracy (mean±standard deviation) in the target domain for the six domain adaptation tasks from the Office-31 dataset are reported in Table 1 and Table 2 with different label ambiguity levels. Table 1 reports the comparison results with a label ambiguity level indicated by $q = 0.2$. From the table, we can observe that the proposed PAPLUDA not only produces the best results on all the six domain adaptation tasks, but also outperforms the comparison methods with substantial performance gains. For example, the proposed approach outperforms the best comparison methods by 20.03%, 15.81%, 15.49%, and 8.87% on the domain adaptation tasks of D → W, A → W, W → D, and A → D, respectively. Such superior performance demonstrates the effectiveness of the proposed PAPLUDA approach in addressing the PLUDA problem.

Table 2 reports the comparison results in a more challenging setting with a higher label ambiguity level of $q = 0.4$. We can see the proposed approach again consistently outperforms all the other methods in this setting. Moreover, the performance gains achieved by the proposed approach over the other comparison methods are very remarkable. For example, PAPLUDA outperforms the best comparison methods by 25.52%, 23.11%, 25.31%, and 12.36% on the domain adaptation tasks of D → W, A → W, W → D, and A → D respectively, and it yields an average performance gain of 18% over the best comparison method, GearNet$_{TCL}$. Compared with the experimental results reported in Table 1, the performance of each comparison method reported in Table 2 in general degrades due to the higher noise level (i.e., label ambiguity level). However, the performance gains achieved by the proposed method over the best comparison methods improve as the noise level increases from $q = 0.2$ to $q = 0.4$. In particular, the average performance gain achieved by PAPLUDA over the best comparison method, GearNet$_{TCL}$, increases from 13.81% to 18% when $q$ increases from 0.2 to 0.4. These results demonstrate the effectiveness of the proposed PAPLUDA method in handling high level label ambiguities.

In summary, the experimental results on the Office-31 dataset with different noise levels validate the effectiveness of the proposed PAPLUDA method in dealing with the challenging PLUDA problem.

### 4.2.2 RESULTS ON OFFICE-HOME

The comparison results on the twelve domain adaptation tasks constructed from the Office-Home dataset are reported in Table 3. From the table, we can observe the follows: (1) Compared with all the other five domain adaptation methods, the proposed approach achieves superior performance across all the twelve tasks. Moreover, its performance gains are very notable. For example, the proposed approach outperforms the best comparison methods by 14.49%, 14.05% , and 12.79% on the tasks of Pr → Cl, Cl → Pr, and Ar → Pr, respectively. (2) Out of the total 60 comparison cases over 5 comparison methods and 12 domain adaption tasks, the proposed approach outperforms all the comparison methods consistently, which is very remarkable given the fact that different comparison methods have different strengths on various classification tasks. These results on the Office-Home dataset again validate the efficacy of the proposed method in addressing the PLUDA problem.

Table 4: Results (%) of the ablation study on multiple tasks.

| Ablation Variant | D $\rightarrow$ A ($q = 0.2$) | D $\rightarrow$ A ($q = 0.4$) | Ar $\rightarrow$ Pr ($q = 0.2$) | Rw $\rightarrow$ Pr ($q = 0.2$) |
|---|---|---|---|---|
| Full Model | 50.07 | 45.28 | 46.78 | 72.30 |
| M-w/o-$\mathcal{L}_{\text{inter}}$ | 42.40 | 36.29 | 44.60 | 69.53 |
| M-w/o–$\mathcal{L}_{\text{reg}}$ | 38.42 | 37.41 | 45.32 | 71.12 |
| M-w/o-soft | 29.87 | 17.65 | 21.17 | 39.92 |
| M-w/o-$F'$ | 47.02 | 37.02 | 44.93 | 71.18 |
| M-w/o-target | 34.28 | 32.70 | 40.52 | 65.30 |
| CLS-source | 19.55 | 10.33 | 13.22 | 32.88 |

## 4.3 ABLATION STUDY

The proposed approach has three main components: soft label disambiguation, inter-domain class-prototype alignment, and teacher-student model based contrastive regularization. To investigate the impact of these components on addressing the PLUDA problem, we conduct an ablation study to compare the proposed full model with the following ablation variants: (1) M-w/o-$\mathcal{L}_{\text{inter}}$, which drops the inter-domain class-prototype alignment loss from the full model by setting $\alpha = 0$; (2) M-w/o-$\mathcal{L}_{\text{reg}}$, which drops the contrastive regularization loss by setting $\beta = 0$; (3) M-w/o-soft, which drops the moving average based soft label disambiguation/update by setting $\lambda = 0$; (4) M-w/o-$F'$, which drops the teacher network $F'$ and uses the student network $F$ instead; (5) M-w/o-target, which drops the target domain from training by setting $\alpha = 0$ and removing the regularization loss $\mathcal{L}_{\text{reg}}^t$ on the target domain; and (6) CLS-source, which is a baseline that only uses a standard classification loss in the source domain by taking the given candidate labels as the prediction targets.

We conduct the ablation experiments on several domain adaptation tasks selected from the two datasets. The comparison results are reported in Table 4. Compared to the full model, we can see that all the ablation variants have notable performance degradation in different degrees. M-w/o-$\mathcal{L}_{\text{inter}}$ has performance drops of over 7.5% on the two D$\rightarrow$ A tasks with different noise levels and over 2.1% on the Ar$\rightarrow$Pr and Rw$\rightarrow$Pr tasks. These performance degradations suggest the inter-domain class-prototype alignment loss is important for the proposed method in bridging the cross-domain discrepancy. M-w/o-$\mathcal{L}_{\text{reg}}$ also demonstrates similar performance degradations, which validates the contribution of the contrastive regularization loss to the proposed approach. It is worth noting that the variant M-w/o-soft produces substantial performance degradation across all the tasks, with performance drops between 20.2% and 32.38%. These results suggest the proposed soft label disambiguation is critical for handling the label noise in PLUDA. The performance gap between the full model and the variant M-w/o-$F'$ suggests that the teacher model is useful for the soft label update and the contrastive regularization and hence contributes to the proposed model. By dropping the target domain from training, M-w/o-target produces inferior performance than the M-w/o-$\mathcal{L}_{\text{inter}}$ variant. This demonstrates the importance of exploiting the target domain data for domain adaptation, and further validates the contribution of the inter-domain class-prototype alignment loss and the teacher-student model based contrastive regularization loss. Moreover, all the five ablation variants mentioned above—M-w/o-$\mathcal{L}_{\text{inter}}$, M-w/o-$\mathcal{L}_{\text{reg}}$, M-w/o-soft, M-w/o-$F'$, and M-w/o-target—outperform the ablation baseline CLS-source that tackles neither the PLL challenge nor the UDA challenge. Overall, this ablation study demonstrates all the proposed components are essential for the proposed PAPLUDA approach, which effectively integrates these parts into a coherent model for tackling the entangled PLL and UDA challenges in the PLUDA problem.

## 5 CONCLUSION

In this paper, we proposed a new challenging problem called partial label unsupervised domain adaptation (PLUDA), which addresses the entangled PLL and UDA challenges simultaneously. We proposed a novel prototype alignment based PLUDA approach called PAPLUDA. It conducts soft label disambiguation in both the source and target domains and minimizes the cross-domain discrepancy by performing inter-domain class-prototype alignment. In addition, a teacher-student model based contrastive regularization is deployed to enhance prediction stability and improve the class-prototypes in both domains, thus consequently helping the inter-domain class-prototype alignment to facilitate domain adaptation. Extensive experimental results demonstrate the proposed PAPLUDA achieves state-of-the-art performance on benchmark datasets.

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

## A APPENDIX

### A.1 TRAINING ALGORITHM

The training algorithm for the proposed PAPLUDA approach is presented in Algorithm 1.

---

**Algorithm 1** Training Algorithm of PAPLUDA

---

**Input**: $D_S$: partial-label source dataset; $D_T$: unlabeled target dataset;
initial PAPLUDA model: $F, F', G, H$; $\alpha, \beta$: trade-off hyperparameters.
**One epoch training:**
 1: **for** iter = 1: iterations **do**
 2:   Sample a mini-batch of samples $B_s = \{(\mathbf{x}_i^s, \mathbf{y}_i^s)\}_{i=1}^{m_s}$ from $D_S$ and
    sample a mini-batch of samples $B_t = \{\mathbf{x}_i^t\}_{i=1}^{m_t}$ from $D_T$.
 3:   Perform soft label disambiguation on $B_s$ according to Eq.(3).
 4:   Compute classification loss $\mathcal{L}_c$ via Eq.(4) by taking the disambiguated soft labels as target.
 5:   Perform soft label disambiguation/update on $B_t$ according to Eq.(6).
 6:   Compute and update the class-prototypes according to Eq.(5) and Eq.(7).
 7:   Compute the inter-domain class-prototype alignment loss $\mathcal{L}_{inter}$ via Eq.(8).
 8:   Compute the teacher-student model based contrastive regularization loss $\mathcal{L}_{reg}$ via Eq.(10).
 9:   Update $\{F, G, H\}$ to minimize the batch-wise training loss in Eq.(11).
10:   Update the teacher model $F'$ according to Eq.(1).
11: **end for**

---

### A.2 PARAMETER SENSITIVITY ANALYSIS

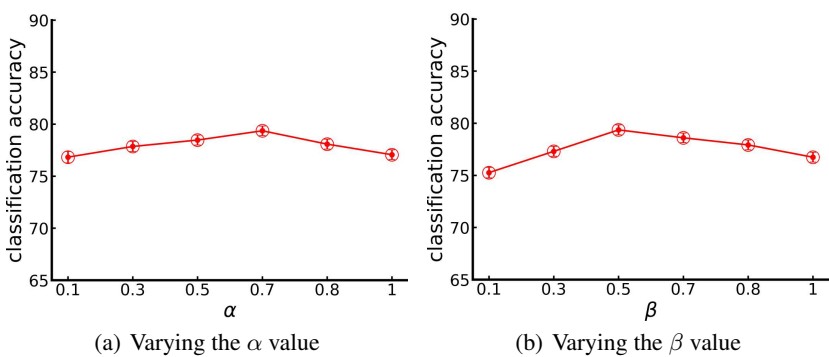

(a) Varying the $\alpha$ value          (b) Varying the $\beta$ value

Figure 2: Test results of the proposed PAPLUDA approach with different $\alpha$ and $\beta$ values on the PLUDA task A $\rightarrow$ W ($q = 0.2$) from Office-31.

We investigate the impact of the trade-off parameters $\alpha$ and $\beta$ on the performance of the proposed approach by conducting experiments on the PLUDA task A $\rightarrow$ W ($q = 0.2$) with different $\alpha$ and $\beta$ values from $\{0.1, 0.3, 0.5, 0.7, 0.8, 1\}$. We first set $\beta = 0.5$ while varying the $\alpha$ value, and then set $\alpha = 0.7$ while varying the $\beta$ value. Note that a larger $\alpha$ value gives greater emphasis on the inter-domain class-prototype alignment loss, and a larger $\beta$ value emphasizes more on the contrastive regularization loss. The test results are presented in Figure 2. We can see that when $\alpha$ is very small, the performance is relatively poor. As $\alpha$ increases, the performance of the approach improves, indicating that the inter-domain class-prototype alignment loss is helpful. However, when $\alpha$ is too large (>0.7), the performance degrades as the inter-domain class-prototype alignment loss gradually dominates. This is reasonable since the inter-domain class-prototype alignment loss is designed to help the model training rather than dominate the learning procedure. The results for different $\beta$ values demonstrate a similar pattern, which indicates the contrastive regularization loss is a useful auxiliary loss for the proposed approach.

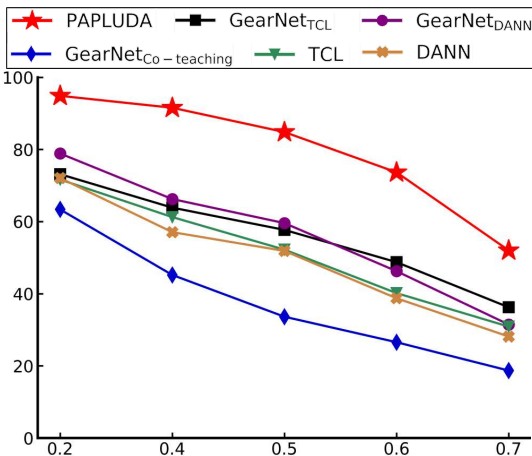

Figure 3: Test results of PAPLUDA with different label noise levels ($q$ values) on the W $\rightarrow$ D task from the Office-31 dataset.

Table 5: Test accuracy (mean±std) comparison on the partial-label Office-Home dataset with label ambiguity level $q = 0.1$. The best result in each row is highlighted in bold.

| Tasks | DANN | TCL | GearNet$_{Co-teach.}$ | GearNet$_{DANN}$ | GearNet$_{TCL}$ | PAPLUDA(Ours) |
|---|---|---|---|---|---|---|
| Ar → CI | 22.04±1.27 | 25.66±1.47 | 20.64±0.52 | 23.67±1.16 | 26.63±0.92 | **37.06±0.95** |
| Ar → Pr | 32.99±1.17 | 38.54±1.31 | 31.20±0.74 | 38.77±1.29 | 40.87±1.11 | **55.33±1.08** |
| Ar → Rw | 41.31±2.17 | 51.55±0.76 | 40.87±0.77 | 45.34±1.32 | 53.86±1.65 | **62.49±1.13** |
| CI → Ar | 28.58±0.71 | 33.21±1.64 | 22.50±0.86 | 33.92±0.81 | 36.62±1.66 | **44.00±0.78** |
| CI → Pr | 29.33±1.17 | 36.70±1.49 | 33.38±0.77 | 35.21±1.22 | 39.27±1.10 | **53.14±1.11** |
| CI → Rw | 41.50±1.49 | 41.40±1.37 | 36.31±1.11 | 47.98±0.63 | 44.18±1.16 | **55.46±0.79** |
| Pr → Ar | 28.42±0.55 | 35.31±0.74 | 24.88±1.37 | 34.01±0.96 | 37.85±1.43 | **45.51±0.91** |
| Pr → CI | 20.93±1.84 | 25.62±1.88 | 17.67±1.28 | 22.86±1.17 | 26.98±1.51 | **40.61±1.12** |
| Pr → Rw | 47.59±1.18 | 54.38±0.91 | 44.12±0.85 | 53.33±1.36 | 58.13±1.25 | **67.40±1.26** |
| Rw → Ar | 39.04±1.15 | 48.88±1.25 | 34.67±1.08 | 42.13±0.82 | 49.64±1.53 | **58.39±0.86** |
| Rw → CI | 25.97±0.98 | 31.47±1.92 | 20.39±1.26 | 27.76±1.17 | 34.22±1.41 | **45.28±1.18** |
| Rw → Pr | 54.85±1.69 | 59.26±1.23 | 56.18±1.28 | 59.51±1.38 | 64.55±1.27 | **72.74±1.21** |
| Average | 34.38±1.28 | 40.16±1.37 | 31.90±0.99 | 38.71±1.11 | 42.48±1.33 | **53.11±1.03** |

## A.3 IMPACT OF LABEL NOISE LEVEL

We also conduct experiments with different label noise levels (i.e., multiple label ambiguity levels—$q$ values) on the domain adaptation task of W $\rightarrow$ D. The comparison results are presented in Figure 3, From the figure, we can see that with the increase of the noise level $q$, the performance of all the comparison methods largely degrades. Nevertheless, the proposed PAPLUDA consistently outperforms all the other comparison methods even in the most challenging case with a large noise level $q = 0.7$, while the other comparison methods produce very poor performance with high noise levels. Moreover, the proposed PAPLUDA maintains large performance gains over the other methods across the range of noise levels. This study further demonstrates the effectiveness of PAPLUDA.

## A.4 ADDITIONAL EXPERIMENTAL RESULTS

To further validate the effectiveness of the proposed approach, we present additional experimental results on Office-Home with a small noise level $q = 0.1$ in Table 5. From the table, we can see that the proposed method produces the best results across all the tasks. Comparing the results in Table 3 and Table 5, we can see that with a smaller $q$ value, the performance of all the comparison methods improves in general, but the proposed PAPLUDA nevertheless maintains notable performance gains over the other methods in both settings.

