# OpenReview forum: "Partial Label Unsupervised Domain Adaptation with Class-Prototype Alignment"
_ICLR.cc/2023/Conference — ICLR 2023 poster_

### Official Review · Reviewer_TPqJ · 2022-10-24

**Confidence:** 4
**Correctness:** 2
**Technical Novelty And Significance:** 4
**Empirical Novelty And Significance:** 3
**Recommendation:** 6

**Clarity, Quality, Novelty And Reproducibility:**

<Clarity>

- This paper is well-organized.
- Several minor concerns:
	- In this paper, "=" sometimes means substitution, but it also means equiality in somewhere else. This is a bit confusing.
	- At the first paragraph in Section 3, Bt should comprise x^t, not x^s.

<Quality>

I have some concerns as described in <Weakness>.

<Novelty>

The problem setting PLUDA proposed in this paper is new and interesting. The proposed method seems to be carefully designed to solve this problem. Importantly, it is simple enough, which will be a good baseline for future studies in the community.

<Reproducibility>

The implementation details seem to be sufficiently provided in the paper.


**Strength And Weaknesses:**


<Strength>

- The new problem setting PLUDA proposed in this paper is interesting and should be practically important in some ML applications.
- The pseudo-label refinement and class-prototype alignment are well designed. Their design seems simple but reasonable.
- The proposed method performs substantially well in the experiments.
- This paper is well-organized.

<Weakness>

- It is not clear to see why we need L_reg in the proposed method. The authors state "We expect this prediction consistency regularization loss in both domains can help induce similar feature representations across domains for the classification task" in Section 3.3, but is there any theory or empirical evidence? Since this regularization loss is computed for each domain, it should not explicitly induce similar representations across domains. In addition, L_reg is computed in the output space differently from standard contrastive learning such as SimCLR, which should not directly regularize the feature representation.
- In the experiments, it would be better to include "source-only" performance, which is actually the performance of standard partial-label learning, in the comparison. Since the experimental setup is new, showing the performance of naive baselines itself should be important to highlight the necessity of PLUDA methods.


**Summary Of The Paper:**

In this paper, the authors formalize a new problem setting, called partial label unsupervised domain adaptation (PLUDA), and propose its solution that jointly conducts partial label learning via pseudo-labeling and domain adaptation by class-prototype alignment.

**Summary Of The Review:**

The problem setting tackled in this paper is new and interesting. However, I have several concerns on the design of the proposed solution and its evaluation. I vote for "weak reject."

---

After rebuttal, I updated my score from 5 to 6.

---

### Official Review · Reviewer_jab8 · 2022-10-24

**Confidence:** 4
**Correctness:** 3
**Technical Novelty And Significance:** 3
**Empirical Novelty And Significance:** 3
**Recommendation:** 8

**Clarity, Quality, Novelty And Reproducibility:**

This is a well-written paper overall. The PLUDA problem that integrates both PLL and UDA problems is novel and interesting, which has been demonstrated the existing methods cannot be used to address the new formalized learning scenario. The proposed method consists of multiple components that address the critical issues in PLL and UDA with the help of a teacher-student model, which has no difficulty in understanding and following it.

**Strength And Weaknesses:**

Strengths:
1、This paper considers a feature distribution discrepancy problem between the training and test data in partial label learning which is always ignored in the existing PLL methods. The paper formalizes this new learning scenario as PLUDA problem that addressing both PLL and UDA problems in a unified framework is novel.
2、Exploiting the prototype as the representative of each class to bridge the gap between the source and target domain is promising.
3、Comprehensive experiments with different noise level show the effectiveness of the proposed method in noise label disambiguation and feature distribution discrepancy.

Weaknesses:
1、Some abbreviations are unclear, for example: in Figure 1, the expression ‘EMA’ is not mentioned in the paper. I guess it is an update operation for teacher model’s parameter, is not it?
2、In section 2, the author discuss previous approaches including their weakness that cannot be used in PLUDA problem. But it is not clear which type of existing methods’ weakness the author tries to tackle in the paper.
3、The author uses the different method to achieve the label disambiguation and pseudo label refinement in the source and target domain respectively. The author tries to achieve the same goal but use different strategy, why?
4、The key issues of PLUDA can be lied in the feature distribution discrepancy and irrelevant noise label. From the ablation study, we can see that dropping the pseudo-label refinement gets a largest performance degradation, while dropping the class-prototype alignment loss which is also important to the framework gets less performance drop, why?
5、The problem considered in the paper is novel, but I wonder can we use the off-the-shelf methods to address the new PLUDA problem? Or use the combination of existing PLL and UDA methods? Why?
6、The feature distribution discrepancy is the core of PLUDA, can you further explain how you can align the feature distribution across domains by the prototype alignment?

**Summary Of The Paper:**

This work formalizes a new learning scenario called partial label unsupervised domain adaptation (PLUDA) which comprises both PLL and PLUDA problems. A novel PAPLUDA approach is proposed to address the challenging PLUDA problem by three components: classification loss with soft label disambiguation, inter-domain class-prototype alignment, and teacher-student model based contrastive regularization, which mutually benefit from each other to enhance the PLUDA learning. Experiments validate the effectiveness of the proposed method in dealing with the new PLUDA problem.

**Summary Of The Review:**

This paper is intriguing and contributes to the filed of partial label learning.

---

### Official Review · Reviewer_UB1N · 2022-10-25

**Confidence:** 5
**Correctness:** 4
**Technical Novelty And Significance:** 4
**Empirical Novelty And Significance:** 3
**Recommendation:** 8

**Clarity, Quality, Novelty And Reproducibility:**

This paper is written clearly and easy to follow. The idea of considering the feature distribution discrepancy problem in PLL is novel and exploit the class prototype to address the feature distribution discrepancy is also interesting.

**Strength And Weaknesses:**

Strengths:

1-The domain discrepancy problem is widely existing in many real-world learning applications. This paper explores the problem of discrepancy between training and test data in PLL which is formalized as a challenging PLUDA problem and proposes a novel PAPLUDA framework to address it.

2-The idea of using class prototype to minimize the discrepancy across both domains is interesting.

3-The experiment results demonstrate the effectiveness of the proposed method in addressing both PLL and UDA problems in a unified framework.

Weaknesses:

1-In this paper, the author integrates both PLL and UDA problems and tries to address them in a unified framework. What motivates you to formalize such a new learning scenario?

2-As you stated, GearNet can also address the noise label and UDA problem, why it can’t be used to address the PLUDA problem? Can we modify it for addressing the PLUDA problem? and what’s the difference between WSDA and PLUDA problems.

3-In the approach section, the author uses a one-hot label indicator vector \hat y_i^s for soft label disambiguation in the source domain, but uses the outputs of the teacher model to update label probability vector p_i^t in the target domain, why?

4-Have you tried to perform the label disambiguation in the source domain and update the label probability vector in the target domain by the same way like Eq.(3) or Eq.(6)? I wonder the different but similar label disambiguation strategy can get a better performance than using the same way?

5-What motivates you deploy the inter-domain class prototype alignment term to alleviate the discrepancy between the source and target domains? Why it works?

6-The class prototype can be obtained in batch-wise by Eq.(5) directly, can you further explain why you update in Eq.(7)?


**Summary Of The Paper:**

This paper proposes an unsupervised domain adaptation framework PAPLUDA under the partial label learning scenario which is formalized as a new problem called partial label unsupervised domain adaptation (PLUDA). The proposed PAPLUDA method disambiguates the irrelevant label noise with the help of a teacher-student model and minimizes the discrepancy between the source and target domains by the inter-domain class prototype alignment. Experimental results validate the effectiveness of PAPLUDA in addressing PLUDA problem.

**Summary Of The Review:**

This is an interesting paper that formalizes a new learning scenario and addresses both challenging problems in a unified framework. The approach is novel and technically solid.

---

### Official Review · Reviewer_uDYu · 2022-10-25

**Confidence:** 3
**Correctness:** 3
**Technical Novelty And Significance:** 3
**Empirical Novelty And Significance:** 3
**Recommendation:** 6

**Clarity, Quality, Novelty And Reproducibility:**

* Clarity: The paper is well-written and easy to follow.
* Novelty and Quality: Please see the strength and weaknesses part.
* Reproducibility: The code is not available. The authors should discuss the neural network architectures and hyperparameters in more detail.

**Strength And Weaknesses:**

## Strong points:
* The motivation for their newly introduced problem is clear and reasonable.
* The Related work is well-written which helps to position this paper well in the related literature.
* The authors also did a good job of elaborating on their proposals and providing the intuition behind each component.
* Their method, PAPLUDA, shows a clear advantage over comparison methods in the new setting.
* The ablation studies are sufficient to further strengthen their claims.

## Weak points:
* For the method part, In Section 3, a lot of claims are made without an appropriate reference. For instance, “We deploy a teacher-student model as the classification model due to its robustness against noisy labels for stable prediction.” Why is this claim true? In addition, it is unclear whether some techniques are newly introduced in this paper or have already existed before. For example, the teacher-student models’ design, moving average soft label, class-prototype learning with moving average pseudo-label, and dynamic dictionary, to name a few. Although this is an empirical paper, the authors’ claims still need to be backed by appropriate references to prior works.
* For the experiment part, while PAPLUDA illustrates strong performance on datasets with high ambiguity, there is a tendency that it cannot outperform baselines when q is small and it approaches 0.

## Questions:
* Is gamma in (3) and (6) the same parameter?
* How hyperparameters (coefficients for moving average)  are chosen in Section 4.3? Why are some of them fixed and one is adaptive?

## Minor comments:
* There are some typos and grammatical errors. For example: “they are restricted to the liner model”, “no one irrelevant label is chosen, we randomly pick a irrelevant label”.

**Summary Of The Paper:**

This paper proposes a novel problem by combining the settings of PLL and UDA. To tackle this challenging setting, they also propose a novel approach named PAPLUDA which consists of three loss components: classification loss with soft label disambiguation; inter-domain class-prototype alignment loss, and teacher-student contrastive regularization loss. To demonstrate the effectiveness of their proposed method, they perform experiments on two synthesized datasets which are generated from two popular DA datasets, Office-31 and Office-Home. In all experiments, PAPLUDA outperforms baselines by a significant margin. In addition, ablation studies are conducted to verify the importance of each component in their method and its sensitivity to different choices of hyperparameters.

**Summary Of The Review:**

Overall the contributions of this paper are two-fold: a novel problem and an efficient solution. The efficiency of their method is backed by experimental results and detailed ablation studies.

---

### Decision · Program_Chairs · 2023-01-20

**Decision:**

Accept: poster

**Justification For Why Not Higher Score:**

On the one hand the work appears good and novel.
On the other hand, some presentations issues have been raised with remarks on the experiments. I also find that the scope of work has some limitations.


**Justification For Why Not Lower Score:**

The paper studies a novel framework and provides an efficient method and the reviewers were unanimous for acceptance.

**Metareview: Summary, Strengths And Weaknesses:**

This paper proposes to formalize a new problem called partial label unsupervised domain adaptation (PLUDA) which can be seen as a unified integration between Partial Label Learning and Unsupervised Domain Adaptation. A novel prototype alignment method is provided in this context. This method is efficient and supported by a large experimental study.

Strengths:
-a nice unified framework to address a novel problem
-an interesting solution
-good experimental evaluation

Weaknesses:
-some presentations issues have been raised, notably key differences with state of the art approaches.
-experiments could be improved in different ways.

During rebuttal, authors have provided multiple answers to the main weak points.

During discussion, reviewers were unanimous to acknowledge that the work is interesting and that the paper should be accepted.

I propose then acceptance.



**Note From Pc:**

if the above contains the word "oral" or "spotlight" please see: "oral" presentation means -> notable-top-5% and "spotlight" means -> notable-top-25%. As stated in our emails, we are disassociating presentation type from AC recommendations